# A Probabilistic Model for Discriminative and Neuro-Symbolic Semi-Supervised Learning

## Abstract

Strong progress has been achieved in semi-supervised learning (SSL) by combining several underlying methods, some that pertain to properties of the data distribution $p(\mathbf{x})$, others to the model outputs $p(\mathbf{y}|x)$, e.g. minimising the entropy of unlabelled predictions. Focusing on the latter, we fill a gap in the standard text by introducing a probabilistic model for *discriminative* semi-supervised learning, mirroring the classical generative model. Several SSL methods are theoretically explained by our model as inducing (approximate) strong priors over parameters of $p(\mathbf{y}|x)$. Applying this same probabilistic model to tasks in which labels represent binary attributes, we also theoretically justify a family of *neuro-symbolic* SSL approaches, taking a step towards bridging the divide between statistical learning and logical reasoning.

## 1 Introduction

In semi-supervised learning (SSL), a mapping is learned that predicts labels $y$ for data points $x$ from a dataset of labelled pairs $(x^l, y^l)$ and unlabelled $x^u$. SSL is of practical importance since unlabelled data are often cheaper to acquire and/or more abundant than labelled data. For unlabelled data to help predict labels, the distribution of $x$ must contain information relevant to the prediction (Chapelle et al., 2006; Zhu & Goldberg, 2009). State-of-the-art SSL algorithms (e.g. Berthelot et al., 2019b;a) combine underlying methods, including some that leverage properties of the distribution $p(\mathbf{x})$, and others that rely on the label distribution $p(\mathbf{y}|x)$. The latter include *entropy minimisation* (Grandvalet & Bengio, 2005), *mutual exclusivity* (Sajjadi et al., 2016a; Xu et al., 2018) and *pseudo-labelling* (Lee, 2013), which add functions of unlabelled data predictions to a typical discriminative supervised loss function. Whilst these methods each have their own rationale, we propose a formal probabilistic model that unifies them as a family of *discriminative* semi-supervised learning (DSSL) methods.

Neuro-symbolic learning (NSL) is a broad field that looks to combine logical reasoning and statistical machine learning, e.g. neural networks. Approaches often introduce neural networks into a logical framework (Manhaeve et al., 2018), or logic into statistical learning models (Rocktäschel et al., 2015). Several works combine NSL with semi-supervised learning (Xu et al., 2018; van Krieken et al., 2019) but lack rigorous justification. We show that our probabilistic model for discriminative SSL extends to the case where label components obey logical rules, theoretically justifying neuro-symbolic SSL approaches that augment a supervised loss function with a function based on logical constraints.

Central to this work are ground truth parameters $\{\theta^x\}_{x \in \mathcal{X}}$ of the distributions $p(\mathbf{y}|x)$, as predicted by models such as neural networks. For example, $\theta^x$ may be a multinomial parameter vector specifying the distribution over all labels associated with a given $x$. Since each data point $x$ has a specific label distribution defined by $\theta^x$, sampling from $p(\mathbf{x})$ induces an implicit distribution over parameters, $p(\theta)$. If known, the distribution $p(\theta)$ serves as a prior over all model predictions, $\tilde{\theta}^x$: for labelled samples it may provide little additional information, but for unlabelled data may allow predictions to be evaluated and the model improved. As such, $p(\theta)$ provides a potential basis for semi-supervised learning. We show that, in practice, $p(\theta)$ can avoid much of the complexity of $p(\mathbf{x})$ and have a concise analytical form known *a priori*. In principle, $p(\theta)$ can also be estimated from the parameters learned for labelled data (fitting the intuition that predictions for unlabelled data should be consistent with those of labelled data). We refer to SSL methods that rely on $p(\theta)$ as *discriminative* and formalise them with a hierarchical probabilistic model, analogous to that for generative approaches. Recent results (Berthelot et al., 2019b;a) demonstrate that discriminative SSL is orthogonal and complementary to methods that rely on $p(\mathbf{x})$, such as *data augmentation* and *consistency regularisation* (Sajjadi et al., 2016b; Laine & Aila, 2017; Tarvainen & Valpola, 2017; Miyato et al., 2018).

We consider the explicit form of $p(\theta)$ in classification with mutually exclusive classes, i.e. where each $x$ only ever pairs with a single $y$ and $y|x$ is *deterministic*. By comparison of their loss functions, the SSL methods mentioned (entropy minimisation, mutual exclusivity and pseudo-labelling) can be seen to impose continuous relaxations of the resulting prior $p(\theta)$ and are thus unified under our probabilistic model for discriminative SSL. We then consider classification with binary vector labels, e.g. representing concurrent image features or allowed chess board configurations, where only certain labels/attribute combinations may be *valid*, e.g. according to rules of the game or the laws of nature. Analysing the structure of $p(\theta)$ here, again assuming $y|x$ is deterministic, we show that logical rules between attributes define its support. As such, SSL approaches that use *fuzzy logic* (or similar) to add logical rules into the loss function (e.g. Xu et al., 2018; van Krieken et al., 2019) can be seen as approximating a continuous relaxation of $p(\theta)$ and so also fall under our probabilistic model for discriminative SSL. Our key contributions are:

- to provide a probabilistic model for *discriminative* semi-supervised learning, comparable to that for classical generative methods, contributing to current theoretical understanding of SSL;
- to consider the analytical form of the distribution over parameters $p(\theta)$, by which we explain several SSL methods, including entropy minimisation as used in state-of-art SSL models; and
- to show that our probabilistic model also unifies neuro-symbolic SSL in which logical rules over attributes are incorporated (by fuzzy logic or similar) to regularise the loss function, providing firm theoretical justification for this means of integrating 'connectionist' and 'symbolic' methods.

## 2 BACKGROUND AND RELATED WORK

Notation: $x_i^l \in \boldsymbol{X}^l$, $y_i^l \in \boldsymbol{Y}^l$ are labelled data pairs, $i \in \{1...N_l\}$; $x_j^u \in \boldsymbol{X}^u$, $y_j^u \in \boldsymbol{Y}^u$ are unlabelled data samples and their (unknown) labels, $j \in \{1...N_u\}$; $\mathcal{X}, \mathcal{Y}$ are domains of $x$ and $y$; x, y are random variables of which $x$, $y$ are realisations. $\theta^x$ parameterises the distribution $p(\mathrm{y}|x)$, and is a realisation of a random variable $\theta$. To clarify: for each $x$, an associated parameter $\theta^x$ defines a distribution over associated label(s) $y|x$; and $p(\theta)$ is a distribution over all such parameters.

### 2.1 SEMI-SUPERVISED LEARNING

Semi-supervised learning is a well established field, described by a number of surveys and taxonomies (Seeger, 2006; Zhu & Goldberg, 2009; Chapelle et al., 2006; van Engelen & Hoos, 2020). SSL methods have been categorised by how they adapt supervised learning algorithms (van Engelen & Hoos, 2020); or their assumptions (Chapelle et al., 2006), e.g. that data of each class form a cluster/manifold, or that data of different classes are separated by low density regions. It has been proposed that all such assumptions are variations of *clustering* (van Engelen & Hoos, 2020). Whilst 'clustering' itself is not well defined (Estivill-Castro, 2002), from a probabilistic perspective this suggests that SSL methods assume $p(\mathrm{x})$ to be a *mixture* of conditional distributions that are distinguishable by some property, e.g. connected dense regions. This satisfies the condition that for unlabelled $x$ to help in learning to predict $y$ from $x$, the distribution of $x$ must contain information relevant to the prediction (Chapelle et al., 2006; Zhu & Goldberg, 2009). In this work, we distinguish SSL methods by whether they rely on direct properties of $p(\mathrm{x})$, or on properties that manifest in $p(\theta)$, the distribution over parameters of $p(\mathrm{y}|x; \theta^x)$, for $x \sim p(\mathrm{x})$. State-of-art models (Berthelot et al., 2019b;a) combine methods of both types.

A canonical SSL method that relies on explicit assumptions of $p(\mathrm{x})$ is the classical generative model:

$$p(\boldsymbol{X}^l, \boldsymbol{Y}^l, \boldsymbol{X}^u) = \int_{\psi, \pi} p(\psi, \pi) p(\boldsymbol{X}^l | \boldsymbol{Y}^l, \psi) p(\boldsymbol{Y}^l | \pi) \underbrace{\sum_{\boldsymbol{Y}^u \in \mathcal{Y}^{N_u}} p(\boldsymbol{X}^u | \boldsymbol{Y}^u, \psi) p(\boldsymbol{Y}^u | \pi)}_{p(\boldsymbol{X}^u | \psi, \pi)} \quad (1)$$

Parameters $\psi, \pi$ of $p(\mathrm{x}|y)$ and $p(\mathrm{y})$ are learned from labelled and unlabelled data, e.g. by the EM algorithm, and predictions $p(y|x) = p(x|y)p(y)/p(x)$ follow by Bayes' rule. Figure 1 *(left)* shows the corresponding graphical model. Whilst generative SSL has an appealing probabilistic rationale, it is rarely used in practice, similarly to its counterpart for fully supervised learning, in large part because $p(\mathrm{x})$ is often complex yet must be accurately described (Grandvalet & Bengio, 2005; Zhu & Goldberg, 2009; Lawrence & Jordan, 2006). However, *properties* of $p(\mathrm{x})$ underpin *data augmentation* and *consistency regularisation* (Sajjadi et al., 2016b; Laine & Aila, 2017; Tarvainen & Valpola, 2017; Miyato et al., 2018), in which true $x$ samples are adjusted, using implicit domain knowledge of $p(\mathrm{x}|y)$, to generate artificial samples of the same class, whether or not that class is known. Other SSL methods consider $p(\mathrm{x})$ in terms of components $p(\mathrm{x}|z)$, where $z$ is a latent representation useful for

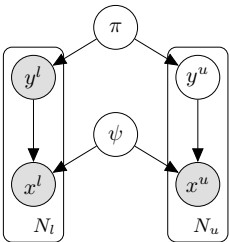 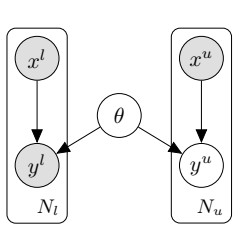 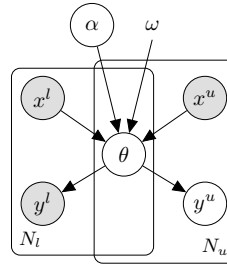

Figure 1: Graphical models for: generative SSL *(left)*; discriminative SSL (previous (Chapelle et al., 2006)) *(centre)*; discriminative SSL (ours) *(right)*. Shading variables are *observed* (else *latent*).

predicting $y$ (Kingma et al., 2014; Rasmus et al., 2015). We focus on a family of SSL methods that add a function of the unlabelled data predictions to a discriminative supervised loss function, e.g.:

- **Entropy minimisation** (Grandvalet & Bengio, 2005) assumes classes are "well separated". As a proxy for class overlap, the entropy of unlabelled data predictions is added to a discriminative supervised loss function $\ell^{sup}$:

$$\ell_{\text{MinEnt}}(\theta) \;=\; \underbrace{-\sum_i \sum_k y_{i,k}^l \log \theta_k^{x_i^l}}_{\ell^{sup}} - \sum_j \sum_k \theta_k^{x_j^u} \log \theta_k^{x_j^u} \qquad (2)$$

- **Mutual exclusivity** (Sajjadi et al., 2016a; Xu et al., 2018) assumes no class overlap, i.e. correct predictions form 'one-hot' vectors. Viewed as vectors of logical variables $z$, such outputs exclusively satisfy the logical formula $\bigvee_k (z_k \bigwedge_{j\neq k} \neg z_j)$. A function based on the formula applies to unlabelled predictions:

$$\ell_{\text{MutExc}}(\theta) \;=\; \ell^{sup} - \sum_j \log \sum_k \theta_k^{x_j^u} \prod_{k'\neq k}(1 - \theta_{k'}^{x_j^u}) \qquad (3)$$

- **Pseudo-labelling** (Lee, 2013) assumes that predicted classes $k_j(t) = \arg\max_k \theta_k^{x_j^u}$ for unlabelled data $x_j^u$ at iteration $t$, are correct (at the time) and treated as labelled data:

$$\ell_{\text{Pseudo}}(\theta, t) \;=\; \ell^{sup} - \sum_j \log \sum_k \mathbb{1}_{k=k_j(t)} \theta_k^{x_j^u} \qquad (4)$$

These methods, though intuitive, lack a probabilistic rationale comparable to that of generative models (Eq. 1). Summing over all labels for unlabelled samples is of little use (Lawrence & Jordan, 2006):

$$p(\boldsymbol{Y}^l | \boldsymbol{X}^l, \boldsymbol{X}^u) = \int_\theta p(\theta) p(\boldsymbol{Y}^l | \boldsymbol{X}^l, \theta) \underbrace{\sum_{\boldsymbol{Y}^u} p(\boldsymbol{Y}^u | \boldsymbol{X}^u, \theta)}_{=1} = \int_\theta p(\theta) p(\boldsymbol{Y}^l | \boldsymbol{X}^l, \theta). \qquad (5)$$

Indeed, under the associated graphical model (Fig. 1 *(centre)*), parameters $\theta$ of $p(\boldsymbol{Y}^l | \boldsymbol{X}^l, \theta)$ are provably independent of $\boldsymbol{X}^u$ (Seeger, 2006; Chapelle et al., 2006). Previous approaches to breaking this independence include introducing additional variables to Gaussian Processes (Lawrence & Jordan, 2006), or an assumption that parameters of $p(\text{y}|x)$ are dependent on those of $p(\text{x})$ (Seeger, 2006). Taking further the (general) assumption of (Seeger, 2006), we provide a probabilistic model for *discriminative* SSL (DSSL), analogous and complementary to that for generative SSL (Eq. 1).

### 2.2 NEURO-SYMBOLIC LEARNING

Neuro-symbolic learning (NSL) aims to bring together statistical machine learning, e.g. neural networks, and logical reasoning (see Garcez et al. (2019) for a summary). Approaches often either introduce statistical methods into a logical framework (e.g. Rocktäschel & Riedel, 2017; Manhaeve et al., 2018); or combine logical rules into statistical learning methods (Rocktäschel et al., 2015; Ding et al., 2018; Marra et al., 2019; van Krieken et al., 2019; Wang et al., 2019). A conceptual framework for NSL (Valiant, 2000; Garcez et al., 2019) places statistical methods within a low-level *perceptual* component that processes raw data (e.g. performing pattern recognition), the output of which feeds a *reasoning* module, e.g. performing logical inference (Fig. 2). This structure surfaces in various works (e.g. Marra et al., 2019; Wang et al., 2019; van Krieken et al., 2019; Dai et al., 2019), in some cases taking explicit analytical form. Marra et al. (2019) propose a 2-step graphical model (their Fig. 1) comprising a neural network and a "semantic layer", however logical constraints are later introduced as a design choice (their Eq. 2), whereas in our work they are a natural way to parameterise a probability distribution. The graphical model for SSL of van Krieken et al. (2019) (their Fig. 1) includes a neural network component and a logic-based prior. However, knowledge base

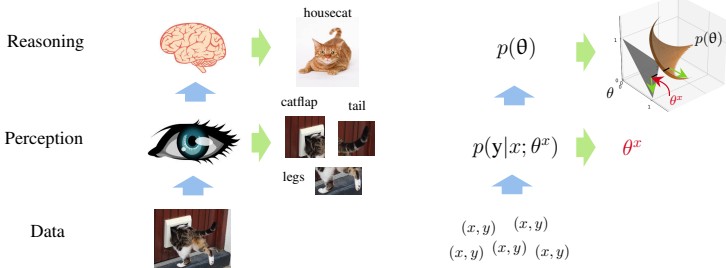

Figure 2: A general framework for neuro-symbolic learning combining statistical learning (perception) and logical rules (reasoning) (Valiant, 2000; Garcez et al., 2019). We draw an analogy to our probabilistic model for discriminative SSL (§3), in which $p(\theta)$ can be defined with logical rules (§5).

rules directly influence labels ($y$) of unlabelled data (only), whereas in our model, rules govern the parameters ($\theta^x$) of all label distributions, i.e. $p(\mathbf{y}|x;\theta^x), \forall x$. Where van Krieken et al. (2019) view probabilities as "continuous relaxations" of logical rules, we show such rules can be used to define the support of the prior $p(\theta)$ in a hierarchical probabilistic model for DSSL, which therefore provides a theoretical basis for neuro-symbolic semi-supervised learning. We note that many other works consider related latent variable models (e.g. Mei et al. (2014) implement logical rules as constraints in a quasi-variational Bayesian approach) or structured label spaces (see e.g. Zhu & Goldberg (2009) for a summary), however we restrict the scope of this review to SSL applications.

## 3    A PROBABILISTIC MODEL FOR DISCRIMINATIVE SSL

Label(s) $y \in \mathcal{Y}$ that occur with a given $x$ can be viewed as samples drawn from $p(\mathbf{y}|x;\theta^x)$, a distribution over the label space with parameter $\theta^x$. For example, in $k$-class classification $p(\mathbf{y}|x)$ is a multinomial distribution over classes, fully defined by a mean parameter $\theta^x \in \Delta^k$, on the simplex in $\mathbb{R}^k$. Every $x \in \mathcal{X}$ has an associated label distribution and so corresponds to a single ground truth parameter $\theta^x$ (in some domain $\Theta$). Thus, there exists a well defined (deterministic) function $f: \mathcal{X} \to \Theta$, $f(x) = \theta^x$. Predictive models, e.g. neural networks, typically learn to approximate $f$: given $x$, they output $\tilde{\theta}^x$, an estimate of $\theta^x$. Note that the label $y$ itself is not predicted, e.g. in the $k$-class classification example, if $x$ occurs with multiple distinct labels across the dataset, their mean $\theta^x = \mathbb{E}[\mathbf{y}|x]$ is learned. Since each $x$ corresponds to a parameter $\theta^x$, sampling $x \sim p(\mathbf{x})$ induces an implicit distribution over parameters $p(\theta)$. In the $k$-class classification example, $p(\theta)$ is a distribution over mean parameters defined on the simplex $\Delta^k$ (e.g. a Dirichlet distribution). Importantly, for any model learning to predict $\theta^x$, $p(\theta)$ serves as a prior distribution over its expected outputs. For a labelled data point, $p(\theta)$ may add little information further to the label, however for unlabelled data, $p(\theta)$ provides a way to evaluate a prediction $\tilde{\theta}^{x^u}$ and so train the model, i.e. by updating it to increase $p(\tilde{\theta}^{x^u})$. We will show (Sec. 4) that under a particular assumption, the analytical form of $p(\theta)$ is known *a priori*. In general, the empirical distribution of predictions for labelled data $p(\tilde{\theta}^{x^l})$ might sufficiently approximate $p(\theta)$. Formalising, let: $\theta^{\mathbf{X}^l} = \{\theta^{x^l}\}_{x^l \in \mathbf{X}^l}$ be the set of parameters of $p(\mathbf{y}|x^l)$ for all $x^l \in \mathbf{X}^l$; and $\theta^{\mathbf{X}^u}$ be defined similarly. Treating $\theta^x$ as a latent random variable with hierarchical prior distribution $p(\theta|\alpha)$, parameterised by $\alpha$, the conditional distribution of the data factorises (analogously to Eq. 5) as:

$$p(\mathbf{Y}^l|\mathbf{X}^l, \mathbf{X}^u) = \int_{\alpha,\theta^{\mathbf{X}},\tilde{\theta}^{\mathbf{X}}} p(\alpha)p(\mathbf{Y}^l|\tilde{\theta}^{\mathbf{X}^l})p(\tilde{\theta}^{\mathbf{X}^l}|\theta^{\mathbf{X}^l})p(\theta^{\mathbf{X}^l}|\alpha) \underbrace{\sum_{\mathbf{Y}^u} p(\mathbf{Y}^u|\tilde{\theta}^{\mathbf{X}^u})}_{=1} p(\tilde{\theta}^{\mathbf{X}^u}|\theta^{\mathbf{X}^u})p(\theta^{\mathbf{X}^u}|\alpha)$$

$$\approx \int_{\alpha,\tilde{\theta}^{\mathbf{X}}} p(\alpha)\, p(\mathbf{Y}^l|\tilde{\theta}^{\mathbf{X}^l})p(\tilde{\theta}^{\mathbf{X}^l}|\alpha)\, p(\tilde{\theta}^{\mathbf{X}^u}|\alpha)\,, \tag{6}$$

where $\tilde{\theta}^x \doteq f_\omega(x)$ represent estimates of (ground truth) $\theta^x$, and $f_\omega : \mathcal{X} \to \Theta$ is a family of functions with weights $\omega$, e.g. a neural network. We replace $p(\mathbf{Y}|\mathbf{X}, \theta^{\mathbf{X}})$ by $p(\mathbf{Y}|\theta^{\mathbf{X}})$ as $\theta^x$ fully defines $p(\mathbf{y}|x)$. The distribution $p(\tilde{\theta}^x|\theta^x)$, over predictions given ground truth parameters, reflects model accuracy (conceptually a *noise* or *error* model) and is expected to vary over $\mathcal{X}$. For *labelled* data, on which the model is trained, we assume (in row 2) that $\tilde{\theta}^x$ closely approximates $\theta^x$, i.e. $p(\tilde{\theta}^x|\theta^x) \approx \delta_{\theta^x - \tilde{\theta}^x}$. For *unlabelled* data, $p(\tilde{\theta}^x|\theta^x)$ is unknown but assumed to increase as predictions approach the true parameter. $p(\tilde{\theta}^x|\alpha) = \int_{\theta^x} p(\tilde{\theta}^x|\theta^x)p(\theta^x|\alpha)$ can be interpreted as a *relaxation* of the prior applied to predictions (a perspective we take going forwards), with equality to the prior in the limiting case $p(\tilde{\theta}^x|\theta^x) = \delta_{\theta^x - \tilde{\theta}^x}$. Fig. 1 (*right*) shows the corresponding graphical model. Taken together, the relationship $f_\omega(x) = \tilde{\theta}^x$, the prior $p(\theta|\alpha)$ and the assumed closeness between $\theta^x$ and $\tilde{\theta}^x$, break the

independence noted previously (Sec. 2). Without $f_\omega$, a sample $x^u$ reveals nothing of $p(\mathrm{y}|x^u; \theta^{x^u})$; without $p(\tilde{\theta}^x|\alpha)$, predictions can be made but not evaluated or improved. Interpreting terms of Eq. 6:

- $p(\boldsymbol{Y}^l | \tilde{\theta}^{\boldsymbol{X}^l})$ encourages labelled predictions $\tilde{\theta}^{x^l}$ to approximate parameters of $p(\mathrm{y}|x^l)$;
- $p(\tilde{\theta}^{\boldsymbol{X}^l} | \alpha)$ allows $\alpha$ to capture the distribution over $\tilde{\theta}^{x^l}$, e.g. to approximate $p(\theta|\alpha)$; and
- $p(\tilde{\theta}^{\boldsymbol{X}^u} | \alpha)$ allows predictions $\tilde{\theta}^{x^u}$ on unlabelled data to be evaluated under prior knowledge of $p(\theta|\alpha)$, or from its approximation learned from labelled data (as above).

Maximum *a posteriori* estimates of $\theta^x$ are given by optimising Eq. 6, e.g. in $K$-class classification by minimising the following objective with respect to $\omega$ (classes indexed by $k$, recall $\tilde{\theta}^x \doteq f_\omega(x)$):

$$\ell_{DSSL}(\theta) \ = -\sum_i \sum_k y^l_{i,k} \log \tilde{\theta}^{x^l_i}_k - \sum_i \log p(\tilde{\theta}^{x^l_i}|\alpha) - \sum_j \log p(\tilde{\theta}^{x^u_j}|\alpha) \tag{7}$$

The $p(\theta|\alpha)$ terms can be interpreted as *regularising* a supervised learning model. However, unlike typical regularisation, e.g. $\ell_1$, $\ell_2$, here it applies to model *outputs* $\tilde{\theta}^x$ not weights $\omega$. Fundamentally, $p(\theta)$ provides a relationship between data samples that enables SSL, as an alternative to $p(\mathrm{x})$.

A natural question arises: given that SSL has fewer $y$ than $x$ by definition, why consider SSL methods that depend on $p(\mathrm{y}|x)$ rather than $p(\mathrm{x})$? Fortunately, the two options are not mutually exclusive, but rather orthogonal and can be combined, as in recent approaches (Berthelot et al., 2019b;a). Furthermore, the structure of $p(\theta)$ is often far simpler than that of $p(\mathrm{x})$ and may be known *a priori*, thus applying DSSL can be straightforward. We analyse the form of $p(\theta)$ in several cases in Sec. 4.

To relate discriminative and generative SSL, we highlight the inherent symmetry between them. Restating the joint distributions behind Eq.s 1 and 6 (see Appendix B for details) as:

$$p(\boldsymbol{Y}^l, \boldsymbol{X}^l, \boldsymbol{X}^u) = \int_{\pi, \psi^{\boldsymbol{Y}}} p(\pi) p(\boldsymbol{X}^l|\psi^{\boldsymbol{Y}^l}) p(\psi^{\boldsymbol{Y}^l}, \boldsymbol{Y}^l|\pi) \sum_{\boldsymbol{Y}^u} p(\boldsymbol{X}^u|\psi^{\boldsymbol{Y}^u}) p(\psi^{\boldsymbol{Y}^u}, \boldsymbol{Y}^u|\pi) \quad \text{[Gen.]}$$

$$p(\boldsymbol{Y}^l, \boldsymbol{X}^l, \boldsymbol{X}^u) = \int_{\alpha, \theta^{\boldsymbol{X}}} p(\alpha) p(\boldsymbol{Y}^l|\theta^{\boldsymbol{X}^l}) p(\theta^{\boldsymbol{X}^l}, \boldsymbol{X}^l|\alpha) \sum_{\boldsymbol{Y}^u} p(\boldsymbol{Y}^u|\theta^{\boldsymbol{X}^u}) p(\theta^{\boldsymbol{X}^u}, \boldsymbol{X}^u|\alpha) \quad \text{[Disc.]}$$

reflects a similar hierarchical structure in which one element of the data ($y$ in the former, $x$ in the latter) acts to 'index' a distribution over the other (see superscript to $\psi$ or $\theta$, resp.).

To have a little understanding of $p(\theta)$, we note that $f(x) = \theta^x$ (assumed differentiable) gives a relationship $p(\theta) = |\boldsymbol{J}| p(\mathrm{x})$, for $\boldsymbol{J}$ the Jacobian matrix $\boldsymbol{J}_{i,j} = \frac{\partial x_i}{\partial \theta_j}$. Thus, if $p(\mathrm{x}) = \sum_k p(x|y=k)\pi_k$ is a mixture distribution with class probabilities $\pi_k = p(y=k)$, then $p(\theta)$ is also:

$$p(\theta) = |\boldsymbol{J}| p(\mathrm{x}) = |\boldsymbol{J}| \sum_k p(\mathrm{x}|y=k)\pi_k = \sum_k |\boldsymbol{J}| p(\mathrm{x}|y=k)\pi_k = \sum_k p(\theta|y=k)\pi_k. \qquad \square$$

Thus any cluster/mixture assumption of $p(\mathrm{x})$ applies also to $p(\theta)$; and class conditional distributions $p(\theta|y)$ over ground truth parameters must differ sufficiently for classification to be possible.

## 4 APPLICATIONS OF DISCRIMINATIVE SEMI-SUPERVISED LEARNING

Implementing Eq. 6 requires a description of $p(\theta)$, ideally in analytic form. Such form depends heavily on two properties of the data: (i) the label domain $\mathcal{Y}$ being continuous or discrete; and (ii) $y|x$ being stochastic or deterministic. Further, discrete labels may represent (a) $K$ distinct classes as 'one-hot' vectors $\boldsymbol{y} \in \{\boldsymbol{e}_k\}_{k \in \{1..K\}}$, where each distribution $p(\mathrm{y}|x)$ is parameterised by $\theta^x \in \Delta^K$ (the simplex), $\theta^x_k = p(\mathrm{y}=k|x)$; or (b) $K$ binary (non-exclusive) features with $\boldsymbol{y} \in \{0, 1\}^K$ (combinations of which give $2^K$ distinct labels), where $\theta^x \in \Delta^{2^K}$, $\theta^x_k = p(\mathbf{y}_k = 1|x)$. Table 1 shows examples for combinations of these factors that determine the form of $p(\theta)$. Italicised cases are discuss in detail.

Table 1: Task and data properties affecting the distribution $p(\theta)$ over parameters of $p(y|x; \theta)$.

| Domain $\mathcal{Y}$ / Map $x \rightarrow y$ | Discrete (classification) | | Continuous (regression) |
|---|---|---|---|
| | Distinct classes | Non-exclusive features | |
| Stochastic | *Mix of Gaussians* | - | - |
| Deterministic | *MNIST, SVHN CIFAR, Imagenet* | *Animals w/attributes Sudoku completion Knowledge Base completion* | Image-to-image translation |

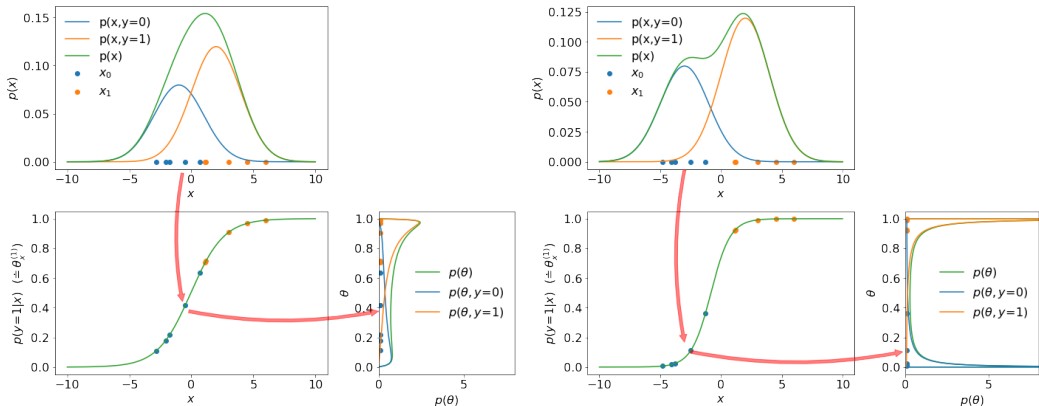

Figure 3: The distribution $p(\theta)$ for a mix of 2 univariate Gaussians (varying class separation).

**Stochastic classification, distinct classes** ($y \in \{e_k\}_k, \theta^x \in \Delta^K$): To see how $p(\mathrm{x})$ and $p(\theta)$ can relate, we consider a mixture of two 1-dimensional equivariant Gaussians: $p(\mathrm{x}) = \sum_k \pi_k \, p(\mathrm{x}|y=k)$, $k \in \{0,1\}$, where $x|y=k \sim \mathcal{N}(\mu_k, \sigma^2)$. Here, $p(\theta)$ can be derived in closed form (see Appendix A):

$$p(\theta) = \sum_{k=0}^1 \pi_k \underbrace{\sqrt{\frac{\sigma^2}{2\pi}} \frac{1}{|\mu_1 - \mu_0|} \exp\{a(\log \tfrac{\theta_1}{\theta_0})^2 + (b_k - 1)\log \theta_1 + (-b_k - 1)\log \theta_0 + c_k\}}_{p(\theta|y=k)}$$

for coefficients $a, b_k, c_k$. As expected (Sec. 3), $p(\theta)$ is a mixture with the same class probabilities as $p(\mathrm{x})$. Fig. 3 shows plots of $p(\mathrm{x})$ and $p(\theta)$ for different separation of class means $\mu_k$, illustrating how the former determines the latter. Just as parameters of $p(\mathrm{x})$ could be revised if a set of unlabelled data samples $x^u$ failed to fit the model, their predictions $\tilde{\theta}^x$ together with $p(\theta)$ offer a similar opportunity. As class means diverge, class overlap reduces and $p(\theta)$ tends towards a limiting *discrete* distribution. Note that, while here both $p(\mathrm{x})$ and $p(\theta)$ are known analytically, this is not typically the case and $p(\mathrm{x})$ may be highly complex, however $p(\theta)$ may still have a concise analytical form.

**Deterministic classification, distinct classes** ($\boldsymbol{y} \in \{e_k\}_k, \theta^x \in \{e_k\}_k$): Often in classification, each $x$ associates (materially) with only one class label $y$, i.e. no other label $y' \neq y$ appears with that $x$; thus classes are mutually exclusive and $y|x$ is *deterministic*. This applies in popular image classification datasets, e.g. MNIST, CIFAR and ImageNet. Where so, $\theta_k^x = p(y=k|x) \in \{0,1\}$ and ground truth parameters (*as well* perhaps as labels) form one-hot vectors $\theta^x \in \{e_k\}_k \subset \Delta^K$. It follows that each class conditional $p(\theta|y=k)$ is (approximately) a delta function $\delta_{\theta - e_k}$ at a vertex of the simplex $\Delta^K$:

$$p(\theta) = \sum_k p(y=k)\, p(\theta|y=k) \approx \sum_k \pi_k \, \delta_{\theta - e_k}. \tag{8}$$

To clarify, for any $x \in \mathcal{X}$, the corresponding parameter $\theta^x$ is always one-hot (the '1' indicating the single corresponding label $y$), ruling out stochastic parameters that imply the same $x$ can have multiple labels. Note that, irrespective of the complexity of $p(\mathrm{x})$, $p(\theta)$ is defined concisely. However, this distribution is discontinuous and lacks support over almost all of $\Delta^K$, i.e. $p(\theta) = 0$ for any $\theta \neq e_k$, making it unsuitable for gradient-based learning methods. However, a continuous approximation to $p(\theta)$ is obtained by *relaxing* each delta component to a suitable function over $\Delta^K$. Such relaxation can be interpreted as estimating a *noise* or *error* model $p(\tilde{\theta}^x|\theta^x)$ of predictions given true parameters (see Sec 3). From Eqs 2, 3 and 4 and Fig. 4, the unlabelled loss components of the SSL methods entropy minimisation (Grandvalet & Bengio, 2005), mutual exclusivity (Sajjadi et al., 2016a; Xu et al., 2018) and pseudo-labelling (Lee, 2013) can be seen to impose (un-normalised) continuous relaxations $\hat{p}(\theta)$ of the discrete $p(\theta)$. (Note: such $\hat{p}(\theta)$ need not be normalised in practice since a weighting term in the loss function renders any proportionality constant irrelevant.) We thus theoretically unify these methods under the probabilistic model for discriminative SSL (Eqs. 6, 7).

**Deterministic classification, non-exclusive features** ($\boldsymbol{y} \in \{0,1\}^K, \theta \in \{0,1\}^K$): In some classification tasks, label vectors $\boldsymbol{y} \in \{0,1\}^K$ represent multiple ($K$) binary attributes of $x$, e.g. features present in an image, a solution to Sudoku, or the relations connecting subject and object entities in a knowledge base. As in those examples, $\boldsymbol{y}|x$ can be deterministic. Where so, for a given $x_*$ and its (unique) label $\boldsymbol{y}_*$, the conditional distribution $p(\mathbf{y}|x_*)$ equates to the indicator function $\mathbb{1}_{\boldsymbol{y} - \boldsymbol{y}_*}$, as parameterised by $\theta^{x_*} = \boldsymbol{y}_*$. Thus, all (true) parameters $\theta^x$ must be at vertices of the simplex $\{0,1\}^K \subset \Delta^{2^K}$ (analogous to one-hot vectors previously). It follows that $p(\theta|\boldsymbol{y}) = \delta_{\theta - \boldsymbol{y}}$ and so $p(\theta) = \sum_{\boldsymbol{y}} \pi_{\boldsymbol{y}} \, \delta_{\theta - \boldsymbol{y}}$ is a weighted sum of point probability masses at $\theta \in \{0,1\}^K$. A continuous

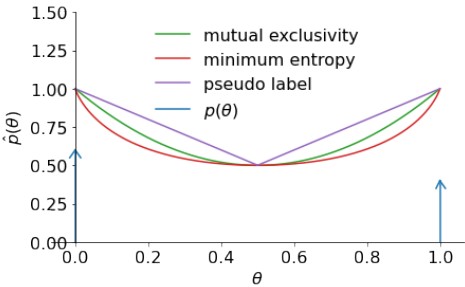

Figure 4: Unsupervised loss components of entropy minimisation (Eq. 2), mutual exclusivity (Eq. 3) and pseudo-labelling (Eq. 4) (exponentiated for comparison to probabilities), seen as continuous relaxations $\hat{p}(\theta)$ of the discrete distribution $p(\theta)$, for deterministic $y|x$ with distinct classes.

relaxation of $p(\theta)$ is again required for gradient based learning. The case becomes more interesting when considering logical relationships that can exist between attributes (Sec. 5). Note that any $\theta \in \{0, 1\}^K \subset \Delta^{2^K}$ in the support of $p(\theta)$ ($2^K$ points in a continuous space) corresponds one-to-one with a label $\boldsymbol{y} \in \{0, 1\}^K$. As such, the distribution $p(\theta)$ could potentially be learned from *unpaired labels* $\boldsymbol{y} \sim p(\boldsymbol{y})$, a variation of typical SSL (we leave this direction to future work).

## 5  NEURO-SYMBOLIC SEMI-SUPERVISED LEARNING

In classification with non-exclusive binary features, certain feature combinations may be impossible, e.g. an animal having both legs and fins, three kings on a chess board, or knowledge base entities being related by *capital_city_of* but not *city_in*. Where so, the support of $p(\boldsymbol{y}|x)$ for any $x$ is confined to a data-specific set of *valid* labels $\mathbb{V}$, a subset of all *plausible* labels $\mathbb{P} = \{0, 1\}^K$, i.e. $p(\boldsymbol{y}|x) = 0$, $\forall \boldsymbol{y} \in \mathbb{P} \backslash \mathbb{V}$. If $\boldsymbol{y}|x$ is *deterministic*, there is a 1-1 correspondence between $\boldsymbol{y}$ and $\theta \in \{0, 1\}^K$ (Sec. 4), and we use $\mathbb{V}, \mathbb{P}$ to refer to both labels $\boldsymbol{y}$ and parameters $\theta$ that are valid or plausible (resp.). Thus:

$$p(\theta|\alpha) \;=\; \sum_{\boldsymbol{y} \in \mathbb{V}} p(\boldsymbol{y}) p(\theta|\boldsymbol{y}) \;=\; \sum_{\boldsymbol{y} \in \mathbb{V}} \pi_{\boldsymbol{y}} \delta_{\theta - \boldsymbol{y}} \,, \tag{9}$$

where $\alpha = \{\mathbb{V}, \Pi_{\mathbb{V}}\}$ and $\Pi_{\mathbb{V}} = \{\pi_{\boldsymbol{y}} = p(\boldsymbol{y})\}_{\boldsymbol{y} \in \mathbb{V}}$ are marginal label probabilities. (Note that Eq, 9 also holds for any 'larger' set $\mathbb{V}'$, where $\mathbb{V} \subseteq \mathbb{V}' \subseteq \mathbb{P}$.) As in the examples mentioned, the set of valid labels $\mathbb{V}$ may be constrained, even defined, by a set of rules, e.g. mutual exclusivity of certain attributes, rules of a game, or relationships between entity relations. Importantly, if a set of rules constrain $\mathbb{V}$, Eq. 9 shows that they constrain the *support* of $p(\theta)$, directly connecting them to the distribution used in discriminative semi-supervised learning (Eqs 6, 7). This is appealing since logical rules possess certainty (*cf* the uncertain generalisation of statistical models, e.g. neural networks) and their universality may allow a large set $\mathbb{V}$ to be defined relatively succinctly. To focus on $p(\theta)$'s support, we drop $\pi_{\boldsymbol{y}}$ and consider probability mass (replacing $\delta_{\theta_k - c}$ with Kronecker delta $\delta_{\theta_k c}$), to define:

$$s(\theta) \;=\; \sum_{\boldsymbol{y} \in \mathbb{V}} \delta_{\theta \boldsymbol{y}} \;=\; \sum_{\boldsymbol{y} \in \mathbb{V}} \prod_{k: \boldsymbol{y}_k = 1} \delta_{\theta_k 1} \prod_{k: \boldsymbol{y}_k = 0} \delta_{(1 - \theta_k) 1} \,, \tag{10}$$

where each term in the summation effectively evaluates whether $\theta$ matches a valid label $\boldsymbol{y} \in \mathbb{V}$, i.e. $s(\theta) = 1$ if $\theta \in \mathbb{V}$, else $s(\theta) = 0$. Restricting to plausible $\theta \in \mathbb{P}$ and defining logical variables $\boldsymbol{z}_k \iff (\delta_{\theta_k 1} = 1)$, it can be seen that Eq. 10 is equivalent to a *logical formula* in propositional logic:

$$\bigvee_{\boldsymbol{y} \in \mathbb{V}} \bigwedge_{k:(\boldsymbol{y}_k = 1)} \boldsymbol{z}_k \bigwedge_{k: \neg(\boldsymbol{y}_k = 1)} \neg \boldsymbol{z}_k \,, \tag{11}$$

which evaluates to True $\Leftrightarrow \theta \in \mathbb{V} \Leftrightarrow s(\theta) = 1$. Comparing Eqs. 10 and 11 shows a relationship between logical and mathematical operations common in fuzzy logic and neuro-symbolic literature (e.g. Bergmann, 2008; Serafini & Garcez, 2016; van Krieken et al., 2019). Here, True maps to 1, False to 0, $\wedge$ to multiplication, $\vee$ to addition, and where $\boldsymbol{z}_k$ corresponds to $\delta_{\theta_k 1} = 1$ (a function of $\theta_k$), $\neg \boldsymbol{z}_k$ maps to $\delta_{(1 - \theta_k) 1} = 1$ (the same function applied to $1 - \theta_k$). In fact, for any ($m$-ary) propositional logic operator $\circ(X_1 \ldots X_m)$, e.g. $X_1 \Rightarrow X_2$, several *functional representations* $\rho_\circ : [0, 1]^m \to [0, 1]$ exist, taking binary inputs, corresponding to $X_i \in \{\text{True}, \text{False}\}$, and outputting $\rho_\circ = 1$ if $\circ$ evaluates to True, else 0 (Bergmann, 2008; Marra et al., 2019). The functional representation for a logical formula composed of several logical operators is constructed by compounding the functional representations of its components. Where two logical formulae are equivalent, then their functional representations are equivalent in that each evaluates to 1 *iff* the logical formula is True, else 0. As such, any set of logical rules that define $\mathbb{V}$ are equivalent to Eq. 11, and can be converted to a functional representation equivalent to $s(\theta)$ in Eq. 10, restricted to $\theta \in \mathbb{P}$.

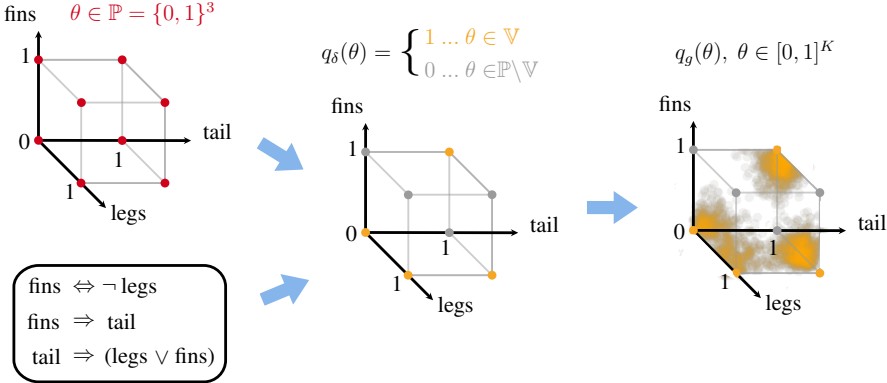

Figure 5: An illustration of the correspondence between logical rules and the support of $p(\theta)$. (*top left*) All *plausible* values of $\theta$ if $y|x$ is deterministic, i.e. $\theta$ restricted to the vertices (Sec. 4). (*bottom left*) an example set of logical rules over label attributes. (*centre*) All *valid* values of $\theta$ under the rules, as encoded in $q_\delta(\theta)$, a function over $\mathbb{P}$ that defines the support of $p(\theta)$. (*right*) $q_g(\theta)$, a relaxation of $q_\delta(\theta)$, defined over $[0,1]^K$, the gradient of which can "guide" unlabelled predictions towards valid $\theta$.

Thus, logical rules can be converted into a function $q_\delta(\theta)$ (defined for $\theta \in \mathbb{P} = \{0,1\}^K$) that evaluates whether a binary vector is in $\mathbb{V}$, the support of $p(\theta)$, i.e. $q_\delta(\theta) = \mathbb{1}_{\theta \in \mathbb{V}}$. Fig. 5 (*left, centre*) gives a simple illustration. As in previous cases, gradient-based learning requires a relaxation of this function defined over the domain of model predictions $[0,1]^K$. This is achieved by replacing the use of $\delta_{\theta_k 1}$ with any function $g(\theta_k) : [0,1] \to [0,1]$, $g(1) = 1$, $g(0) = 0$ (a relaxation of $\delta_{\theta_k 1}$). By choosing $g$ continuous, the resulting $q_g(\theta) : [0,1]^K \to [0,1]$ is continuous and satisfies $q_g(\theta) = s(\theta) = 1$ for valid $\theta \in \mathbb{V}$, and $q_g(\theta) = s(\theta) = 0$ for invalid $\theta \in \mathbb{P} \backslash \mathbb{V}$, providing a continuous relaxation of $p(\theta)$, $\forall \theta \in [0,1]^K$ (Fig. 5, *right*), up to probability weights $\pi_{\boldsymbol{y}}$ (see Appendix C). Thus the distribution $p(\theta)$ required for DSSL (Eqs. 6, 7), can be approximated by a functional representation of logical rules.

In practice, the choice $g(\theta_k) = \theta_k$ from *fuzzy logic* (Bergmann, 2008) is often used (e.g. Serafini & Garcez, 2016; van Krieken et al., 2019; Marra et al., 2019). Applying this choice directly to Eq. 10 gives the *semantic loss* (Xu et al., 2018), which is thus probabilistically justified and unified under the model for discriminative SSL. Under the same DSSL model, $p(\theta)$ can also be learned from the labelled data; justifying the use of logical techniques, such as *abduction* (Wang et al., 2019; Dai et al., 2019), to extract rules consistent with observed labels, i.e. that entail $\mathbb{V}$.

Many works combine functional representations of logical formulae with statistical machine learning. We have shown that such methods are theoretically justified and that logical rules fit naturally into a probabilistic framework, i.e. by defining the support of $p(\theta)$, the distribution necessary for discriminative semi-supervised learning.

## 6 CONCLUSION

In this work, we present a hierarchical probabilistic model for *discriminative* semi-supervised learning, complementing the analogous model for classical generative SSL methods. Central to this model are the parameters $\theta^x$ of distributions $p(y|x; \theta^x)$, as often predicted by neural networks. The distribution $p(\theta)$ over those parameters serves as a prior over the outputs of a predictive model for unlabelled data. Depending on properties of the data, in particular whether $y|x$ is deterministic, the analytical form of $p(\theta)$ may be known *a priori*. Whilst not explored in this paper, the model for DSSL shows that an empirical estimate of $p(\theta)$ might also be learned from labelled data predictions (or indeed unpaired labels). In cases where labels reflect multiple binary attributes, logic relationships may exists between attributes. We show how such rules fit within the same probabilistic model for DSSL, providing a principled means of combining logical reasoning and statistical machine learning. Logical rules can be known *a priori* and imposed, or potentially learned from the data. Our single model for discriminative semi-supervised learning probabilistically justifies and unifies families of methods from the SSL and neuro-symbolic literature, and accords with a general architecture proposed for neuro-symbolic computation (Valiant, 2000; Garcez et al., 2019), comprising low level *perception* and high level *reasoning* modules (Fig 2). In future work, we plan to consider the more complicated case where $y|x$ is stochastic (i.e. combining aleatoric uncertainty); to make more rigorous the notion of a *noise* or *error* model (i.e. capturing epistemic uncertainty), and to extend the principled combination of statistical machine learning and logical reasoning to supervised learning scenarios.

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

## APPENDIX A  DERIVATION OF $p(\theta)$ FOR A MIXTURE OF GAUSSIANS

For a general mixture distribution:

$$\theta_k^x = p(y=k|x) = \sigma\Big( \log \frac{p(x|y=k)\pi_k}{\sum_{k'\neq k} p(x|y=k')\pi_{k'}} \Big);$$

$$\frac{d\theta_k^x}{dx} = \theta_k^x(1-\theta_k^x)\Big( \frac{d}{dx}\log p(x|y=k) - \sum_{k'\neq k} \frac{p(x|y=k')\pi_k}{\sum_{k''\neq k} p(x|y=k'')\pi_{k''}} \frac{d}{dx}\log p(x|y=k') \Big)$$

which, in our particular case, become:

$$\theta_1^x = \sigma\Big( \log \frac{\pi_1}{\pi_0} + \frac{\mu_1-\mu_0}{\sigma^2}x - \frac{1}{2}\big( \frac{\mu_1^2}{\sigma^2} - \frac{\mu_0^2}{\sigma^2} \big) \Big), \qquad \frac{d\theta_k^x}{dx} = \theta_k^x(1-\theta_k^x)\big( \frac{\mu_1^2}{\sigma^2} - \frac{\mu_0^2}{\sigma^2} \big).$$

Rearranging the former gives $x$ in terms of $\theta$. Substituting into $p(\theta) = |\boldsymbol{J}|p(\mathrm{x})$ gives:

$$p(\theta) = \sum_{k=0}^{1} \pi_k \underbrace{\sqrt{\frac{\sigma^2}{2\pi}} \frac{1}{|\mu_1-\mu_0|} \exp\{a(\log \frac{\theta_1}{\theta_0})^2 + (b_k-1)\log\theta_1 + (-b_k-1)\log\theta_0 + c_k\}}_{p(\theta|y=k)}$$

with coefficients: $a = \frac{-\sigma^2}{2(\mu_1-\mu_0)^2}$, $b_k = \frac{\mu_k}{\mu_1-\mu_0} + \frac{\sigma^2}{(\mu_1-\mu_0)^2}\big( \frac{\mu_1^2-\mu_0^2}{\sigma^2} - \log \frac{\pi_1}{\pi_0} \big)$, $c_k = -\frac{(\mu_1-\mu_0)^2 b_k^2}{2\sigma^2}$.

## APPENDIX B  RECONCILING EQUATIONS (1) AND [GEN.]

$$p(\boldsymbol{X}^l, \boldsymbol{Y}^l, \boldsymbol{X}^u) = \int_{\psi,\pi} p(\psi,\pi)p(\boldsymbol{X}^l|\boldsymbol{Y}^l,\psi)p(\boldsymbol{Y}^l|\pi)\sum_{\boldsymbol{Y}^u} p(\boldsymbol{X}^u|\boldsymbol{Y}^u,\psi)p(\boldsymbol{Y}^u|\pi) \tag{1}$$

$$= \int_{\psi,\pi} p(\pi)p(\psi^{\boldsymbol{Y}^l},\psi^{\boldsymbol{Y}^u})p(\boldsymbol{X}^l|\boldsymbol{Y}^l,\psi^{\boldsymbol{Y}^l})p(\boldsymbol{Y}^l|\pi)\sum_{\boldsymbol{Y}^u} p(\boldsymbol{X}^u|\boldsymbol{Y}^u,\psi^{\boldsymbol{Y}^u})p(\boldsymbol{Y}^u|\pi)$$

$$= \int_{\psi,\pi} p(\pi)p(\boldsymbol{X}^l|\boldsymbol{Y}^l,\psi^{\boldsymbol{Y}^l})p(\psi^{\boldsymbol{Y}^l},\boldsymbol{Y}^l|\pi)\sum_{\boldsymbol{Y}^u} p(\boldsymbol{X}^u|\boldsymbol{Y}^u,\psi^{\boldsymbol{Y}^u})p(\psi^{\boldsymbol{Y}^l},\boldsymbol{Y}^u|\pi)$$

$$= \int_{\psi,\pi} p(\pi)p(\boldsymbol{X}^l|\psi^{\boldsymbol{Y}^l})p(\psi^{\boldsymbol{Y}^l},\boldsymbol{Y}^l|\pi)\sum_{\boldsymbol{Y}^u} p(\boldsymbol{X}^u|\psi^{\boldsymbol{Y}^u})p(\psi^{\boldsymbol{Y}^u},\boldsymbol{Y}^u|\pi) \qquad \text{[Gen.]}$$

Explanation: Each term $\psi^{\boldsymbol{Y}}$ parameterises a distribution of the form $p(\boldsymbol{X}|\boldsymbol{Y},\psi^{\boldsymbol{Y}})$. Those distributions are conditional on the labels $\boldsymbol{Y}$, hence we attach that label to the respective parameter to identify the correspondence. Such parameters are referred to collectively as $\psi$ in line 1. Line 2 separates them and identifies where each occurs elsewhere. Since labels $\boldsymbol{Y}$ and their associated parameters $\psi^{\boldsymbol{Y}}$ go hand in hand, they are probabilistically interchangeable: we could think of drawing each label $y$ from a pool of $k$ labels (and the parameter $\psi^y$ comes with it), or draw a parameter $\psi^y$ from a pool of $k$ parameters. This explains the last 2 lines. For clarity, note that in $p(\boldsymbol{X}^u|\boldsymbol{Y}^u,\psi^{\boldsymbol{Y}^u})$, $\boldsymbol{Y}^u$ can be considered redundant, given the parameter of the distribution, the identity of the label of that distribution is irrelevant.

## APPENDIX C  JUSTIFICATION FOR CONSIDERING ONLY THE SUPPORT OF $p(\theta)$

In section 5, we focus on the *support* of $p(\theta)$ defined by the set of valid binary vectors $\mathbb{V}$, ignoring the corresponding class probabilities $p(\boldsymbol{y}) = \pi_{\boldsymbol{y}} \in \Pi_{\mathbb{V}}$. The discriminative SSL methods analysed (see Eqs 2, 3, 4) ignore class weights also. Practical reasons for doing are (i) that they may not be known, and (ii) that unless attributes are independent ($p(\boldsymbol{y}) = \prod_k p(\boldsymbol{y}_k)$), class probabilities do not factorise across dimensions equivalently to the support (Eq. 10). We briefly consider the validity and implications of considering only the support of $p(\theta)$.

**Validity**: Regardless of whether capturing all aspects of $p(\theta)$ is preferable, considering only its support is valid since it is equivalent to assuming a uniform distribution over the support. The resulting approximation to $p(\theta)$ might be considered a "partially-uninformative" prior.

**Implications**: Considering only the support of $p(\theta)$ may be sufficient for the purposes of DSSL, since where $p(\theta)$ provides a prior over unlabelled predictions, it is not used alone. If we had the full (discrete) $p(\theta)$ and it alone were used to predict labels for unlabelled data, a maximum likelihood approach would simply assign the most common label to all unlabelled data points. However, $p(\theta)$ is used in conjunction with a supervised model that learns to approximates $f(x) = \theta^x$ to gives predictions $\tilde{\theta}^x$, taking class probabilities into account. To the extent the model generalises, its predictions on unlabelled data should correlate with (i.e. be close to) their true values $\theta^x$. Therefore a function $q_g(\theta)$, that is a continuous relaxation of the support of $p(\theta)$, helps by guiding predictions $\tilde{\theta}^x$ to nearby valid values of $\theta$, which should to some extent reflect the correct labels. Intuitively, the prior class weights may be useful for data samples where the model is highly uncertain, where the best option may again be to choose the most popular class. For well balanced classes, ignoring $\pi_{\boldsymbol{y}}$ should have little impact.

