# OpenReview forum: "A Probabilistic Model for Discriminative and Neuro-Symbolic Semi-Supervised Learning"
_ICLR.cc/2021/Conference — Reject_

### Official Review · AnonReviewer3 · 2020-10-15
**Discriminative semi-supervised learning**

**Rating:** 7
**Confidence:** 2

**Review:**

The authors introduce a discriminative model for semi-supervised learning for which several existing methods are special cases. In their model, for each data value, there is a distribution from which the label is sampled. Although this distribution is unknown, in their framework the sampling distribution's parameters are approximately produced by a discriminative model such as a neural network trained on the labeled data.

When the labels are vectors of features, the prior distribution of this parameter distribution can be defined so as to enforce logical constraints about which combinations of features are valid. This is a neat direction and in keeping with recent trends to integrate statistical and logical reasoning, although the paper would be strengthened if the authors gave concrete examples of a real-world dataset for which their innovation would be helpful. Likewise for their overall model, it's nice to help unify previous work, but additional discussion of impact or potential applications beyond what's been done would strengthen the paper.

---

> ### Author Response · Authors · 2020-11-22
> **Response to Reviewer 3**
>
> Many thanks for your review. We have updated the paper throughout to make the central idea more accessible and the connection between SSL neuro-symbolic methods more clear. We have also included (Fig. 5) a simple example of how a set of logical rules can translate into an approximation to the distribution p(theta) that enables semi-supervised learning.

---

### Official Review · AnonReviewer2 · 2020-10-19
**promises more than it delivers**

**Rating:** 5
**Confidence:** 3

**Review:**

The paper aims at proposing a theoretical rationale for discriminative semi-supervised learning that is comparable with that of generative models. Moreover,
the paper aims at theoretically justifying a family of neuro-symbolic SSL approaches.
For the first task, the paper states that the proposal justifies entropy minimisation (Grandvalet &
Bengio, 2005), mutual exclusivity (Sajjadi et al., 2016a; Xu et al., 2018) and pseudo-labelling (Lee,
2013).
For the second task, the paper states that the proposal justifies  (Serafini & Garcez, 2016; van Krieken et al., 2019, Marra et al., 2019, Xu et al., 2018).

With respect to the first task, while I agree the the paper provides a justification for mutual exclusivity (Sajjadi et al., 2016a)
and pseudo-labelling (Lee, 2013), I have doubts on entropy minimisation (Grandvalet & Bengio, 2005):
how do you model an entropy with terms of the form $-\sum\log p(\theta|\alpha)$?
The authors also say that previous discriminative SSL lack theoretical justification but (Xu et al., 2018) proves that their loss is a consequence of a
number of assumptions, thus theoretically justifying it.

The paper misunderstood the scope of neuro-symbolic integration (NeSy): it is not the integration of statistics and logic but the integration of
neural networks with (possibly probabilistic) logic. The integration of statistics and logic is well established in the field of
Statistical Relational Learning.  For example, (De Raedt et al., 2007) is not NeSy while (Manhaeve et al., 2018) is.
As such, the paper does not provide "a theoretically principled understanding of integrating ‘connectionist’ and ‘symbolic’ methods." in general, only
for some specific methods (not for example for (Manhaeve et al., 2018)). In fact, the justification is only for NeSy that are based on fuzzy logic,
which is much simpler to manipulate than probabilistic logic as in (Manhaeve et al., 2018). Fuzzy logic is not a probabilistic logic because the
first is truth functional (the value of A and B depends only on the values of A and B) while the latter is not (A and B depends on whether A and B are
independent, mutually exclusive,...). Moreover, (Xu et al., 2018) is more than simply impositive mutual exclusivity as it  also allow a form of probabilistic reasoning by
representing the loss formula as an arithmetic circuit obtained with automated reasoning. So overall I think that the paper claims more than it effectively provides.

The presentation of the paper makes following the exposition unnecessarily difficult.
First, the notation is confusing, with a nonuniform representation
of vectors, which are sometimes bold and sometimes not ($\mathbb{z}$ and $\theta$), while sometimes also scalars are bold ($\mathbb{z}_k$).
Moreover, random variables and their values are also not typographically distinct.
All integrals have the integrating variables as subscripts of the integral symbol instead of as $dxdy$...
In Figure 1 the left and center subfigures are equal, I guess in the left one x and y should be exchanged.
Figure 1 left should represent the model of eq 1 but $\psi$ and $\pi$ are absent from the figure.

In eq. 6, shouldn't the integrating variable be $\tilde{\theta}^X$?
It is not clear how you derive the formulas [Gen.] from eq 1: since Figure 1 does not show parameters
$\psi$ and $\pi$, it is difficult to judge the conditional independences among variables.
The symbol $\Delta^{2^K}$ is confusing since it can also be interpreted as the simplex over a set with $2^K$ elements. Better
$(\Delta^2)^K$.
You say that Figure 3 represents a mix of two univariate Gaussians but the variance of the two Gaussians do not seem equal: if only
the mean changes, then the Gaussians should have the same height.
The paper mentions Table 4 which is absent from the paper.

Minor comments:
"proportionality constant obsolete"->"proportionality constant irrelevant"
A the bottom of page 6, last formula: the subscript of the sum should be $\mathbb{y}$
Page 7 "Note that each $\theta\in\{0,1\}^K\subseteq \Delta^{2^K}$ (from a continuous space): $\theta$ does not seem from a continuous space.
Bibliography: reference Olivier Chapelle, Bernhard Scholkopf, and Alexander Zien. Semi-supervised learning. IEEE
Transactions on Neural Networks, 20(3):542–542, 2009. is a book review, you want to cite the book, not the review
In Jingyi Xu, Zilu Zhang, Tal Friedman, Yitao Liang, and Guy Broeck. A semantic loss function for
deep learning with symbolic knowledge. In International Conference on Machine Learning, 2018.
the last author is Guy van den Broeck


---After reading the other reviews and the authors' comments, I sill think that the paper promises more than it delivers, even if the paper was extensively rewritten as a consequence of many problems in the original version, so I will keep my score.

---

> ### Author Response · Authors · 2020-11-22
> **Response to Reviewer 2**
>
> Many thanks for your review. We address each of your comments below and in the revised paper.
>
> 1. “entropy minimisation (Grandvalet & Bengio, 2005): how do you model an entropy with terms of the form -\sum\log p(\theta | \alpha)?”
>     - Under gradient based optimisation, any strictly convex function over the simplex that is maximal at its vertices (ie for one-hot vectors) serves as a relaxation of the true distribution p(\theta), (a weighted sum of delta functions at simplex vertices). The negative entropy of a multinomial is known to satisfy those requirements. Mathematically: let p(\theta | alpha) \propto \prod_k (\theta_k)^{\theta_k} aligns Eqs 2, 6.
>
> 2. “(Xu et al., 2018) proves that their loss is a consequence of a number of assumptions, thus theoretically justifying it”
>     - Agreed, existing works have separate rationale (which we make more clear), whereas we unify those works under a single probabilistic model. Interesting future work might consider which axioms of Xu et al are satisfied by *any* relaxation of the prior, our g(\theta).
>
> 3. “The paper misunderstood the scope of neuro-symbolic integration”
>     - Understood, we have addressed this.
>
> 4. “the paper does not provide "a theoretically principled understanding of integrating ‘connectionist’ and ‘symbolic’ methods” in general, only for some specific methods"
>     - We did not intend this to read so broadly (emphasis was meant on “*a* theoretically…”), we have reworded to clarify and limit the claim.
>
> 5. “(Xu et al., 2018) is more than mutual exclusivity, it also allow a form of probabilistic reasoning”
>     - Agreed, Xu et al. covers mutual exclusivity (Sec 4) & logical reasoning (Sec 5), so we reference it for both, showing that it can be interpreted under a probabilistic framework and unified with other approaches.
>
> 6. "notation is confusing"
>    - We have revised the notation to be more consistent, in particular (non-italicised) \uptheta is used for the random variable, consistent with other RVs.
>
> 7. "All integrals have the integrating variables as subscripts of the integral symbol instead of as dxdy"
>     - We believe this is fairly common shorthand notation, e.g. Barber, “Bayesian Reasoning & Machine Learning”. However, we will review this for the camera ready version (if necessary).
>
> 8. (i) "Figure 1 left should represent the model of eq 1" (ii) "In eq. 6, shouldn't the integrating variable be \tilde\theta^X?"
>     - Agreed, amended
>
> 9. "It is not clear how you derive the formulas [Gen.] from eq 1"
>     - The rearrangement is now explained in Appendix B.
>
> 10. "The symbol \Delta^{2^K} is confusing since it can also be interpreted as the simplex over a set with 2^K elements"
>     - Without assuming attribute independence, there are 2^K distinct labels and an arbitrary multinomial distribution over them has (2^K) - 1 free parameters. Assuming attribute independence reduces the domain to [0,1]^K hypercube (isomorphic to a region of the simplex).
>
> 11. "Figure 3 the Gaussians should have the same height."
>     - The plots show p(x,y) = p(x|y)p(y) and so are Gaussians weighted by p(y).
>
> 12. (i) "The paper mentions Table 4 which is absent." (ii) "obsolete"->"irrelevant". (iii) “page 6, last formula: subscript should be 𝕪“
>     - Amended.
>
> 13. “Page 7 Note that each \theta \in {0,1}^K \subseteq \Delta^(2^K) (from a continuous space):  does not seem from a continuous space”
>     - Clarified.
>
> 14. "Bibliography: references to Chapelle et al., Xu et al."
>     - Amended.

---

> > ### Comment · AnonReviewer2 · 2020-11-23
> > **answer 1**
> >
> > I dont understand the formula \prod_k (\theta_k)^{\theta_k} , are you sure that it is correct? You exponentiate a parameter vectore by itself?

---

> > > ### Author Response · Authors · 2020-11-23
> > > **response to answer 1**
> > >
> > > Considering the (K-dimensional) prediction $\theta$ (drop superscript here, just for ease of notation) for a single unlabelled data sample $x_j$, the min entropy term (Eq 2) is:
> > >    - $
> > >      -\sum_k \theta_k \log \theta_k
> > > =   -\sum_k  \log \theta_k^{\theta_k}
> > > =   -\log \prod_k\theta_k^{\theta_k}
> > > $
> > >
> > > Equating this to the $-\log p(\theta|\alpha)$ term in Eq (7) shows that $\prod_k\theta_k^{\theta_k}$ describes the implied form of $p(\theta|\alpha)$ (up to proportionality) assumed under min entropy.  Importantly, this function is convex over [0,1] and maximal when each $\theta_k$ is 0 or 1, hence it acts as a relaxation of the "discrete" distribution under the assumption of deterministic $y|x$.

---

### Official Review · AnonReviewer4 · 2020-10-26
**Recommendation to Reject**

**Rating:** 4
**Confidence:** 4

**Review:**

#### Summary

This paper proposes a probabilistic model to describe semi-supervised/unsupervised learning, which is further applied to model neuro-symbolic learning. Comparing to traditional unsupervised/semi-supervised learning formulations, the proposed model imposes a prior on the label distribution instead of input features. When applying this formulation to neuro-symbolic learning, the symbolic part can be regarded as a prior on label space to constrain the learning process. Finally, the authors propose three methods to calculate the loss of violating the symbolic prior constraints on label space.

#### Pros

+ This paper is motivated by a very important problem. Combining statistical learning and symbolic reasoning becomes a trend recently, many algorithms and models have been proposed. However, there lacks a unified theoretical framework to understand this combination fundamentally.
+ Modelling neuro-symbolic learning as a traditional semi-supervised statistical learning problem is an interesting idea. The resulted model integrates symbolic constraints as a prior on label distribution, which is very natural.

#### Cons
- The writing of this paper could be improved. For example, the notation in this paper is sometimes confusing. The meaning of parameter $\theta^x$ is not clearly explained: In section 2, it is explained as a parameter of the conditional distribution $p(\mathrm{y}|x)$, so I took it as the parameters of a classifier $y=h_\theta^x(x)$. But in Section 4 and 5, the authors state that $\mathbf{y}$ and $\theta$ has a 1-1 corresponds in the space of $\{0,1\}^K$, so it seems that $\theta^x$ should be interpreted as pseudo-labels of examples. In section 3, the authors introduce another variable $\tilde{\theta^x}$ as estimations of $\theta^x$, which makes me more confused. Although this is a theoretical paper, I think the authors should improve the notations and provide some simple examples for helping readers understand this framework.
- The idea of using logic rules as constraints (or other non-differentiable constraints) on label space is not a novel idea in statistical learning (Zhang and Zhou, 2013). In section 3, the first category of "deterministic, distinct classes" is multi-class learning; the "deterministic non-exclusive features" is multi-label learning. In both contexts, there are many works talking about constraining the label space with prior knowledge, for example, Cesa-Bianchi et al. (2006) learns multi-label classifier with hierarchical labels; Tsochantaridis et al. (2004) proposes SVM-struct deal with structural output with support vector machines; Mei et al. (2014) introduces logic rules as label constraints within a Bayesian learning framework. The authors have surveyed a wide range of related works in the area of semi-supervised learning and neuro-symbolic learning. However, I think the works I have mentioned above is highly related to this work; the authors should discuss the relationship between this work and those early works in statistical learning.
- Last but not least, this paper lacks experiments to support the authors' claims. Meanwhile, although this paper presents a general framework, it provides little insight into this area. The authors propose three methods to compute the non-differentiable $p(\theta)$, while (B) ignores the relationship between rules (e.g., recursive theories) and treat rules independently; (C) is equivalent to multi-label learning setting or the ECOC method in multi-class learning.

#### Recommendation

Overall, I appreciate that this paper is trying to solve a crucial problem in neuro-symbolic learning, and the idea of viewing this problem as semi-supervised statistical learning is natural and reasonable. However, this paper should include more discussion about related works in statistical learning. Furthermore, it would be better if the authors can include some examples and experiments to demonstrate the idea of the proposed model.

#### Additional questions and comments

- The first to sub-figures in Fig. 1 are the same; the generative model should have arrows from $y$ and $\theta$ to $x$.
- In section 3, first paragraph, line 6: "... outputting $\theta^x$, an estimate of $\theta^x$". The first $\theta^x$ should be $\tilde{\theta^x}$.
- $\alpha$ first appears in Eq. 6 and Fig. 1. However, there is no explanation about what is it.


#### References
- Zhang, Min-Ling, and Zhi-Hua Zhou. "A review on multi-label learning algorithms." IEEE transactions on knowledge and data engineering 26.8 (2013): 1819-1837.
- Cesa-Bianchi, Nicolò, Claudio Gentile, and Luca Zaniboni. "Incremental algorithms for hierarchical classification." Journal of Machine Learning Research 7.Jan (2006): 31-54.
- Tsochantaridis, Ioannis, et al. "Support vector machine learning for interdependent and structured output spaces." Proceedings of the twenty-first international conference on Machine learning. 2004.
- Mei, Shike, Jun Zhu, and Jerry Zhu. "Robust regbayes: Selectively incorporating first-order logic domain knowledge into bayesian models." International Conference on Machine Learning. 2014.

---

> ### Author Response · Authors · 2020-11-22
> **Response to Reviewer 4**
>
> Many thanks for your review. We address each of your comments below and in the revised paper.
> 1. “writing of this paper could be improved”
>     - Secs 1-3 have been re-worded for greater clarity.
> 2. “The meaning of parameter \theta^x is not clearly explained ... the authors should improve the notations and provide some simple examples”
>     - We have made this more clear. p(Y|x) is a distribution over (all) labels for a given sample of x (note: no two distributions p(Y|x_i), p(Y|x_j) need be the same). Each distribution is defined by a parameter \theta^x (e.g. a vector on the simplex in the case of multi-class classification). Since each distribution is specific to some x, we use that x to index (by a superscript) the corresponding parameter.
>     - For clarity: we now never use “parameter” to refer to the “weights” of a model, only ever parameters of a probability distribution (e.g. its mean).
>     - Under a hierarchical Bayesian model, we consider the distribution p(\theta) over those parameters (superscript x is dropped when referring to the random variable, which is now denoted \uptheta for greater clarity).
>     - We denote estimates of parameter (e.g. as output by a model) with \tilde.
>     - Throughout, we focus on the special case where each x only ever occurs with a single y, rather than perhaps appearing with several distinct y_1, y_2, y_3 in the dataset (e.g. as possible in the mixture of Gaussians case). In this case, every distribution p(Y|x) is a delta distribution and so can be parameterised by a single label y  (or its one-hot equivalent), i.e. \theta^x=y  (hence the 1-1 correspondence between \theta^x and y mentioned).
>
> 3. “using logic rules as constraints (or other non-differentiable constraints) on label space is not a novel idea”.
>     - Agreed. Note, we do not introduce logical rules as a constraint. Rather, for a family of existing SSL methods that do, we show that they fall under a unifying probabilistic model for DSSL.
>
> 4. “In [multi-class learning & multi-label learning], there are many works talking about constraining the label space with prior knowledge … the authors should discuss the relationship between this work and those early works in statistical learning”
>     - Thankyou for the references. The paper is updated to acknowledge extensive prior work in related areas. Since our work specifically addresses SSL to unify specific families of methods under a probabilistic model, we restrict our detailed review to works most closely related.
>
> 5. “this paper lacks experiments to support the authors' claims”
>     - Our work is completely theoretic, describing a new theoretical explanation for an approach to SSL that also provides a unifying probabilistic rationale for two families of existing methods including a theoretical link between neuro-symbolic and statistical machine learning methods. We have added an example into the neuro-symbolic section to give clearer insight.
>
> 6. “provides little insight into this area”
>     - We have reworded the paper to make insights more clear.
>
> 7. “include some examples”
>     - We have made the running example of multi-class classification more clear in Sec 3. Figure 3 also aims to give a visual understanding how p(x) corresponds to p(\theta). We have also added an example to the neuro-symbolic section (new Fig. 5).
>
> 8. "The authors propose three methods to compute the non-differentiable ..."
>     - We have removed this section in order to make the text more clear and to include Fig. 5. The reviews have shown that it is most fundamental to make the central idea of the paper more clear, i.e. how the single probabilistic model unifies families of models and those families (from SSL & NSL) with each other.
>
> 9. “The first two sub-figures in Fig. 1 are the same.”
>     - Thanks and apologies. An error in the latex referenced the wrong subfigure.
>
> 10. “The first should be \tilde\theta^x.”
>     - Agreed, amended.
>
> 11. “\alpha ... no explanation about what is it.”
>     - Agreed, amended.

---

> > ### Comment · AnonReviewer4 · 2020-11-24
> > **Feedback after reading the rebuttal**
> >
> > Thank you for the detailed clarification, and I appreciate the authors' revision. However, I still think the presented framework is not distinguished from prior works in SSL and Bayesian learning. The lack of working example and experiment weakens the contribution of this work. Therefore I will keep my rating.

---

> > > ### Author Response · Authors · 2020-11-24
> > > **Response to rebuttal feedback**
> > >
> > > Thank you for your further response. Please note:
> > >
> > > 1. [re: Distinguishing from prior works in SSL] we include 2 pages of Background material, in which we summarise:
> > >      - existing theory behind Semi-Supervised Learning,
> > >      - example SSL methods that are covered by our theoretical model; and
> > >      - relevant Neuro-symbolic approaches, including relevant high-level perspectives from the less recent (e.g. Valiant (2000), see Fig 2) through to very recent (e.g. Garcez at al (2019)).
> > >
> > >    Our work exclusively addresses SSL, providing a theoretical explanation that unifies families of methods from both traditional SSL and neuro-symbolic SSL, hence we devote space to providing sufficient related background for the paper to be broadly self-contained.
> > >
> > >    We carefully distinguish the proposed model for SSL from prior theoretical explanations, showing a novel symmetry with the more familiar generative model for SSL (see equations labelled [Gen] & [Disc], Sec 3).
> > >
> > > 2. [re: Distinguishing from prior works in Bayesian learning] we do not *distinguish* from Bayesian learning as such, since we propose a Bayesian model, but we are unaware of this model being used to explain semi-supervised learning or neuro-symbolic SSL anywhere in the literature (including the several extensive SSL reviews referenced). The only Bayesian explanation for SSL that we are aware of is the generative model with which we draw direct contrast (noted above).
> > >
> > > 3. [re: References mentioned previously] There are many machine learning works with which comparisons might be drawn (e.g. any that involve supervised model with a prior) but it is not possible to discuss all such works. However, as suggested, we acknowledge that a broader backdrop exists. Note that, of the references suggested previously:
> > >    - Mei et al. (2014) impose spike-and-slab-type regularising constraints in a latent variable model that applies approximate inference for unsupervised learning (i.e. LDA topic modelling). The authors motivate their quasi-bayesian approach on the basis that when minimising KL divergence between true & approximate posteriors, "it is often difficult to make sure that the posterior satisfies all domain knowledge constraints". In contrast, we consider a fully Bayesian model in which domain knowledge constraints correspond to mathematical properties of the prior, i.e. restricting its support. Where the prior's support is zero, so too is any posterior's, which the considered MAP approaches maximise. We have expanded the suggested reference to summarise this, but do not include further detail since they do not consider (i) semi-supervised learning, (ii) the effect of y|x being deterministic or (iii) how logical rules directly manifest in the prior.
> > >    - Tsochantaridis et al. (2004) considers various structured output spaces, but focuses on adapting large-margin classifiers (SVMs) to such output spaces (e.g. formulating dual programs) and is inherently non-probabilistic in nature. It is difficult to see how it relates to our probabilistic explanation for semi-supervised learning.
> > >    - Cesa-Bianchi et al. (2006) focus on making "hierarchical classifications" over a taxonomy, devise a loss function specific to taxonomic data and consider its theoretical properties. We find it difficult to make any insightful comparison between that work and our own (we do not focus on hierarchy or any *particular* logical relationship between label attributes).
> > >    - Zhang & Zhou (2014) provide a general review of "multi-label" learning, i.e. where label features may co-occur. A main focus of the paper is to categorising approaches (of the time), primarily by the number of label attributes between which interactions are considered. Reviewed methods include nearest-neighbour, decision tree and SVM approaches. There is little clear relationship to any of our work beyond that both consider multi-label leaning to an extent. We reference them in that regard (as suggested).
> > >
> > > 4. [re: working example & experiments] We have made the running example of multi-class classification more prevalent throughout the theory (Sec 3), to give more clear intuition. We have also included an explicit simple diagram in the neuro-symbolic part (Sec 4) to make this more tangible. So there are *two* working examples. Further, various specific methods (e.g. entropy minimisation, mutual exclusivity, semantic loss) are explained and shown to be instantiations of our general theoretical framework. Each of those works include experiments that demonstrate their relative performance on a number of tasks. As such, there is significant existing empirical evidence that these methods "work", we provide a unifying theoretical basis to give clearer insight why. Similarly with neuro-symbolic SSL, several works show that using logical rules as a regulariser improves performance, but mathematical understanding of how such rules fit with a probabilistically interetable supervised learning loss function has been lacking.

---

### Official Review · AnonReviewer1 · 2020-10-29
**Relevant theme but still preliminary: many ideas but still scattered results.**

**Rating:** 3
**Confidence:** 4

**Review:**

The authors discuss several ideas aimed at improved semi-supervised learning by adopting an appropriate "plate model" with probabilistic content, and then examining various techniques and variants. The theme is relevant but the whole effort seems a bit preliminary: there are many keywords and many discussed techniques, but the whole picture is not clear in terms of concrete contributions (and the provided testing does not clarify whether gains are realized).

The paper contains a somewhat long introduction that in a sense includes Sections 1 and 2, then quickly goes through the proposed Expression (6), and derives consequences that are not always clearly articulated; for instance what is the point of Table 1? Also the connection with neuro-symbolic learning is interesting but it feels a bit too much; why exactly is it needed in this framework? Or is it just an optional add-on? (Besides, for the proposed approach to work I believe more testing is needed.)

A problem: as far as I can tell, Figure 1 left and center are identical. What is the difference?

A problem: is the citation (Van Engelen and Hoos 2020) indeed in the references? I could not find it.

---

> ### Author Response · Authors · 2020-11-22
> **Response to Reviewer 1**
>
> Many thanks for your review, we address each of your comments below and in the revised paper:
>
> 1. “not clear in terms of concrete contributions … connection with neuro-symbolic learning is interesting but feels a bit much; why is it needed in this framework?”
>     - We have revised the paper to be more clear and connected, including adding an example into the neuro-symbolic section.
>     - We show that several discriminative SSL methods can be theoretically justified and unified by a single probabilistic model in which a prior over the parameters of p(y|x) encodes the assumption that y|x is deterministic (each x has only one label). Example: in multiclass classification, such parameters (\theta^x) are vectors on the simplex. Given test input x, a typical classifier can output any simplex vector, whereas the imposed prior confines parameters to the simplex vertices (i.e. one-hot), a continuous relaxation of that “encourages” model predictions to be one-hot.
> The DSSL methods considered each have rationale (e.g. minimising entropy) but we show that *all* approximate the “deterministic” prior and so are unified under one probabilistic model.
>     - Considering *the same* probabilistic model when labels are binary vectors, an equivalent deterministic prior over the parameter space again restricts parameters to its vertices. If logical rules determine attribute combinations, they restrict the prior to “valid” vertices and so can be “naturally” built into the prior in a rigorously justified way.
>     - Summarising: *one probabilistic model* unifies a family of DSSL methods *and* a family of neuro-symbolic SSL methods. We believe it is important to include both to show that connection.
>
> 2. “long introduction … quickly goes through Expression (6)”
>     - Secs 1-3 have been reworded for greater clarity.
>
> 3. “point of Table 1?”
>     - Table 1 shows combinations of data attributes (with examples) that dictate the form of p(\theta), i.e. its specific form in an SSL loss function. We explore several cases (now italicised) in detail. We have made this more clear in the paper.
>
> 4. “Figure 1 left and center identical”
>     - Thanks, amended.
>
> 5. “(Van Engelen and Hoos 2020) in the references?”
>     - Yes, 5th from last, p.10

---

### Decision · Program_Chairs · 2021-01-07
**Final Decision**

**Decision:**

Reject

**Comment:**

Reviewers agree that this is a very promising paper, with an excellent overview of existing techniques for semi-supervised and neuro-symbolic learning. However, reviewers also agree that the paper is not ready. With one more revision for clarity, some limited empirical validation and illustration of the theory, and focus on the essential message, this could become a seminal paper for our understanding of semi-supervised learning. Luckily the reviews provided ample feedback, and the authors should be able to submit a very competitive paper next time around.